


Diversity, distribution and nitrogen use strategies of bacteria in the South China Sea basin
Yuan-Yuan Li[1], Xiao-Huang Chen[1], Peng-Fei Wu[1], Dong-Xu Li[1], Lin Lin[1], Da-Zhi Wang[1,2]*
[1]State Key Laboratory of Marine Environmental Science/College of the Environment and Ecology, Xiamen
University, Xiamen, 361005, China
[2]Key Laboratory of Marine Ecology & Environmental Sciences, China Academy of Sciences, Qingdao,
266071, China
**Correspondence:** Da-Zhi Wang (dzwang@xmu.edu.cn)



**Abstract** The diversity and abundance of bacteria and diazotrophs in the euphotic and aphotic layers of the
South China Sea (SCS) basin were investigated based on high-throughput sequencing of the 16S rRNA and
nifH genes. Bacterial communities in the aphotic layers significantly differed from those at the euphotic
layers, and were characterized by geographical specificities. *Prochlorococcus* and *Alphaproteobacteria* were
abundant in the surface layer, whereas *Gammaproteobacteria* was more common in the aphotic layers.
*Moraxellaceae* was the most abundant group in the aphotic layer in the northern basin of the SCS (nSCS),
while *SAR324*, *SAR202* and *SAR406* occurred mainly in the southern basin of the SCS (sSCS). Diazotrophic
*Alphaproteobacteria* was the predominant group in the SCS basin, whereas Marine Group II *Euryarchaeota*
emerged in the euphotic bottom of both nSCS and sSCS. Abundances of genes encoding amino acid
transporters and ammonium assimilating enzymes were relatively high in the SCS surface and the entire
water column of the sSCS, while expression levels of urea and ammonium transporter-encoding genes were
the highest at the surface of the SEATS site. Iron deficiency-induced gene *IdiA* and urease were highly
expressed at the A2 site. Our results indicated that bacterial communities in the SCS were depth-stratified
and exhibited geographic divergency in the aphotic layers between nSCS and sSCS. Amino acids and
ammonium were the major nitrogen sources for bacteria while urea, ammonia and nitrite played important
roles in regulating cell growth of *Prochlorococcus* in different regions of the SCS.












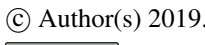



## 1 Introduction

As a key component of the marine ecosystem, bacteria play indispensable roles in the carbon cycling and

regulation of global climate (Pedro ́s-Alio ́, 2006). The diversity and distribution of bacterial communities in

the oceans are influenced by various environmental factors (DeLong et al. 2006; Daryabor et al. 2016).

Nitrogen (N), an essential macronutrient for bacteria, is one of the most important factors regulating

bacterial growth, diversity and distribution in the oceans, especially in the subtropical and tropical

oligotrophic oceans, in which dissolved inorganic N is almost undetectable. Accordingly, bacteria have

evolved diverse strategies to adapt to ambient N deficiency, i.e. up-regulating expressions of high-affinity

transporters or utilizing dissolved organic N (DON), such as urea, amino acids and polyamines (Kent et al.,

2016). In addition, N fixation by diazotrophs represents another important adaptive strategy for bacteria to

survive in N-deficient environments (Montoya et al., 2004). Unicellular and filamentous diazotrophs are

found in the oligotrophic oceans and contribute significantly to N cycling in the global oceans. Therefore,

the elucidation of N utilization strategies of bacteria will advance our understanding of the diversity and

distribution of bacteria in different oceanic regions.

The South China Sea (SCS) is one of the largest marginal seas in the world, characterized by permanently

stratified and oligotrophic waters. N is nearly undetectable in surface waters, and the N deficiency severely

limits bacterial growth and productivity (Wu et al. 2003). Metagenomics of the water column in the South

East Asia Time-series Study (SEATS) site reveals that *Alphaproteobacteria* dominate the surface bacterial

community while *Gammaproteobacteria* thrive in the deep waters (Tseng et al. 2015). *Prochlorococcus* is

the most prevalent autotrophic picoplankton, occurring particularly in summer (Liu et al. 2007; Xie et al.

2018). *SAR11* represents the predominant species in all regions and *Cyanobacteria* proliferate mainly in the

euphotic layer of the north SCS (Jiang et al., 2013). To date, the explorations on N-fixing bacteria in the SCS

basin are still in progress. Using quantitative PCR and molecular cloning methods, *Alphaproteobacteria* and

*Gammaproteobacteria* have been identified as the main diazotrophs in the euphotic zone of the SCS basin,

while *Trichodesmium* and unicellular *Cyanobacteria* exhibit very low abundances (Zhang et al., 2011).

*Gammaproteobacteria* and *Richelia* dominate the diazotrophic communities in the Vietnam Bay and the

Mekong river, respectively (Moisander et al., 2008; Bombar et al., 2011). A pyrosequencing study targeting

the nifH gene reveals that *Gammaproteobacteria* and *Trichodesmium* are the two dominant nifH



phylogenetic groups in the nSCS (Xiao et al. 2015). These studies have substantially improved our understanding of the bacterial diversities in the SCS basin. Nevertheless, most of them are conducted in the nSCS and the coast regions of the sSCS, while almost no effort has been devoted to the sSCS basin. Moreover, previous studies fail to identify key or new unicellular diazotroph species due to the limitations of methodologies.

Isotopic tracing, quantitative PCR and meta-omic approaches have been applied to study the marine N cycle in the SCS and to provide instructive information. *Proteobacteria, Cytophaga-Flavobacteria* and *Cyanobacteria* have been demonstrated to play pivotal roles in nitrate assimilation in the nSCS (Cai and Jiao, 2008). Bacteria are found to exhibit depth-dependent metabolic potentials, and the metabolism of urea and amino acids is more active at the surface of nSCS (Wang et al., 2010; Tseng et al., 2015). In addition, the nSCS basin shows lower $N_2$ fixation activity than the East China Sea and the nSCS shelf (Wu et al., 2018). However, systematic studies on N use strategies of bacterial communities in the SCS basin are still scarce.

The present study examined the diversity and spatial distributions of bacteria and diazotrophs in the SCS basin using high-throughput sequencing of the 16S rRNA and nifH genes. The SEATS site located at the nSCS and the SS1 site located at the sSCS were selected to compare bacterial communities throughout the entire water column, including the surface layer, the deep chlorophyll maximum (DCM) layer, the bottom of the euphotic layer at a depth of 200 m and the oxygen minimum zone (OMZ) at 750 m. PICRUSt predictions and real-time qPCR analysis of major N utilization genes were used to infer the N use strategies of bacterial communities. This study provides insights into the diversity and distribution of bacteria and diazotrophs in complex hydrological environments and nitrogen utilization strategy in the marginal basin regions. It also serves as a pioneer study for the comparison of bacterial N utilization strategies between the nSCS and sSCS regions.

**2 Materials and methods**

**2.1 Sample collection and environmental characteristics**

Bacterial samples were collected from 5$^{th}$ June to 27$^{th}$ June 2017 during the southwest monsoon prevailing period. The SEATS site (18°15'N and 115°30'E) was located at the nSCS, and the other four sites, SS1, A2, B1 and C1, were located at the sSCS (10°-15'N and 110°-120°E) **(Fig. 1; Table 1)**. Seawater at 5 m depth



from sites A2, B1 and C1, and seawater at four different depths from SEATS and SS1 sites (5, 68, 200 and

750 m from SEATS, 5, 105, 200 and 750 m from SS1) were collected using Niskin bottles attached to a

CTD rosette. Approximately 10 L seawater was pre-filtered through a 3 μm pore-size polycarbonate

membrane (47 mm diameter, Millipore) and then retained on a 0.22 μm pore-size polycarbonate membrane.

The membranes were then stored at -80 ℃ on board until use.

Temperature, salinity, depth and dissolved oxygen data were retrieved from the

conductivity-temperature-depth rosette system (CTD, Sea Bird Electronics). Water samples for analysis of

inorganic nutrients ($NO_2^-$+$NO_3^-$, $NO_2^-$, $SiO_3^{2-}$, $PO_4^{3-}$) were filtered through a 0.22 μm pore-size

polycarbonate membrane and then analyzed immediately on board using the automatic continuous AA3 flow

analyzer (Germany) (Fei and Sun, 2011). Sea surface temperature and salinity data were obtained using the

Seabird SBE21 apparatus. Seawater for chlorophyll a (Chla) determination was filtered on a 0.45 μm

pore-size GF/F membrane (Whatman) and then analyzed using the Turner Designs Model 10 fluorometer.

## 2.2 Nucleic acid extraction and reverse transcription

Environmental DNA of each sample was extracted using the FastDNA SPIN Kit (MP Laboratories, Inc.)

following the protocol of the manufacturer. Three biological repeats of environmental RNA were extracted

using the Trizol regent and chloroform, followed by purification using the RNeasy Mini Kit (Qiagen,

Germany) as described by Atshan et al. (2012). Reverse transcriptional experiment was immediately

conducted following the instruction of the QuantiTect Reverse Transcription Kit (Qiagen, Germany). The

extracted DNA and synthetic cDNA samples were stored at -20 ℃.

## 2.3 PCR and sequencing of 16S rRNA and nifH genes

The V3 and V4 regions of 16S rRNA gene in the environmental DNA samples were amplified with

region-specific primers 341F (5'-CCTAYGGGRBGCASCAG-3') and 806R

(5'-GGACTACNNGGGTACTAAT-3') (Yu et al. 2005). Fragments of the nifH gene in all DNA samples

were amplified with specific primers nifH-F (5'-AAAGGYGGWATCG GYAARTCCACCAC-3') and

nifH-R (5'- TTGTTSGCSGCR TACATSGCCATCAT-3') as recommended by Török and Kondorosi (1981).

Routine PCR was carried out using the following thermal cycle: 95 ℃ for 3 min, 27 cycles (35 cycles for





nifH) of 95 ℃ for 30 s, 55 ℃ for 30 s, 72 ℃ for 45 s, and finally 72 ℃ for 10 min. The triplicate PCR
products from each sample were purified by 2% agarose gel electrophoresis and extracted using the AxyPrep
DNA Gel Extraction Kit (Axygen, USA). The purified 16S rRNA amplicons were sequenced using
paired-end sequencing ($2 \times 250$) on the MiSeq platform from Illumina, Inc. Raw reads were de-multiplexed,
quality-filtered using QIIME (v1.9.1) with the criteria as described by Li et al. (2018). The resulting
qualified 16S rRNA sequences were aligned to the Silva database (Release 128, http://www.arb-silva.de),
while the nifH sequences were aligned to the FunGene database under GeneBank (Release 7.3,
http://fungene.cme.msu.edu/). Operational taxonomic units (OTUs) were defined with a percentage
sequence similarity of ⩾ 99% based on the RDP Bayesian classifier algorithm (v2.2). The sequences of
16S rRNA and nifH genes were deposited in GenBank under the BioProject ID PRJNA434503. The
individual accession numbers were SAMN08563407-08563415 for the 16S rRNA samples and
SAMN08563568-08563574 for the nifH samples.

**2.4 Design and validation of *Prochlorococcus* specific primers**
Using the genomes of the sequenced *Prochlorococcus* strains provided by the Cyanobacterial
KnowledgeBase (Peter et al. 2015) as templates, eight group-specific primers targeting N utilizing genes,
including ammonia transporter *(amt1),* urea transporter *(urtA),* amino acid transporter *(AAT),* nitrite
reductase *(nirA),* urease *(ureA),* glutamine synthetase *(GlnA),* ferredoxin-dependent glutamate synthase
*(GltS)* and irondeficiency-induced gene *(IdiA)*, were designed using the online Primer Designing Tool (PDT)
**(Table S1)**. Primers targeting *urtA* and *amt1* genes were referenced from a previous study (Li et al. 2018).
Cyanobacteria-specific primers (16SCF: 5'-GGCAGCAGTGGGGAATTT TC-3' and 16SUR:
5'-GTMTTACCGCGGCTG CTGG-3') were used as internal control genes (Kyoung-Hee et al., 2012) to
minimize sampling or processing differences among samples. To ensure the specificity of the primers, only
hyper-conserved sequences among different *Prochlorococcs* ecotypes were conveyed to the automatic
generation area of PDT. Amplified products for each pair of primers were separated on agarose gel and then
cloned into a T-vector (Takara). At least 35 clones were randomly chosen, fully sequenced and aligned to the
NCBI database. Only the primers that yielded more than 30 positive clones with identity > 90% and E-value
< 0.01 were considered qualified (Bayer et al., 2014; Li et al., 2018).



**2.5 Quantitative real-time PCR assay**

Levels of gene expression were quantified on an ABI 7500 instrument. The qPCR reaction was performed following the protocol of the SYBR Green PreMix Plus Kit (Qiagen, Germany) in a volume of 20 μL. 0.4 μL ROX Reference Dye was added to correct the errors of the fluorescent signals between holes. The thermal cycle conditions were set as follows: preheating at 95 ℃ for 15 s, followed by 40 cycles, with each cycle of heating at 95 ℃ for 15 s and 60 ℃ for 1 min. Relative quantification of target genes was performed by the matched 7500 software (v1.3.1) with the baselines and the cycle threshold (Ct) values set automatically. The relative levels of gene expression were calculated using the $2^{-\Delta\Delta CT}$ method as described by Livak and Schmittgen (2001).

**2.6 Statistical analysis**

The within-habitat diversity (α-diversity) was assessed by the Ace and Shannon indices using the Mothur software (v1.30.1) at a cutoff level of 1%. The between-habitat diversity was assessed by the principal coordinates analysis (PCoA) and the hierarchical clustering analysis based on the Bray-Curtis distance calculated with QIIME Pipeline (Caporaso et al. 2010). The unweighted pair-group with arithmetic mean algorithm was used to build the tree structure. Both the correlations between community structures (revealed from Bray-Curtis distance) and environmental factors, and community diversity estimators with environmental factors, were analyzed by the Mantel test in R software (v3.4.3, vegan package). The Spearman's correlation analysis was performed to assess the correlations between species and environmental factors using the IBM Predictive Analytics Software (PASW) Statistics (v18). All generated coefficients were subjected to the t-test for significance analysis. The heatmaps were generated using the R package "pheatmap".

Based on the 16S rRNA dataset, PICRUSt (v0.9.2) was used to predict the functional contents of the metagenome. The abundances of OTU at 99% identity were standardized by removing the influence of 16S rRNA marker gene on the genome copy numbers to ensure that the OTU abundances accurately reflected the true abundances of the designated organisms. Each OTU was then mapped to the Greengenes database (v13.5) for functional prediction. The resulting functional predictions were assigned to the EggNOG database (v4.0) for all genes. The free online Majorbio I-Sanger Cloud Platform (www.i-sanger.com) was




used for the bioinformatics analysis.

**3 Results**
**3.1 Physicochemical parameters**
The upper mixed water layer of the nSCS near the Luzon Strait was obviously affected by the Kuroshio
Current, characterized by high temperature (30 to 31 ℃) and high salinity (33.5 to 34). The sSCS surface
was dominated by high temperature and sub-high salinity (32.5 to 33.5). The concentrations of $NO_X$ and
$PO_4^{3-}$ were undetectable in the SCS surface while the concentration of $SiO_3^{2-}$ varied between 1 μM and 7 μM.
With increasing water depth, the temperature and concentration of dissolved oxygen decreased rapidly, but
the concentrations of nutrients increased (**Table 1**). The concentration of Chla ranged from 0.08 mg/m$^3$ to
0.25 mg/m$^3$ among the sampling sites. The depth corresponding to maximum Chla concentration was around
68 m in the SEATS site and 105 m in the SS1 site, and the bottom of the euphotic layer was around 200 m.

**3.2 Bacterial community diversity and overall 16S rRNA composition**
In this study, sequencing of the V3 and V4 hypervariable regions of 16S rRNA gene was performed to
characterize the composition and diversity of bacterial communities. Illumina sequencing generated 823,
138 reads in total after quality control. On average, 74, 830 reads were generated per sample with a length of
435 bp per read. With the 99% similarity criteria, a total of 1, 427 different OTUs were obtained from 11
samples with the average OTU number of 582. The coverage for each sample exceeded 99%, indicating that
the selected sequences could indeed represent the bacterial communities of individual samples **(Table S2).**
The bacterial alpha diversities were evaluated by the Ace and Shannon estimators. Bacterial community
richness (Ace) of the surface and DCM layers was lower than that of the deep layers **(Fig. 2A)**. On the
contrary, the within-habitat diversity (Shannon) of bacterial community decreased with increasing water
depth in SEATS but increased in SS1 **(Figure 2C; Table S2)**. To examine the differences of bacterial
community composition between the habitats, OTU and Bray-Curtis distance-based PCoA plot and
hierarchical clustering analysis were further performed. The samples were found to form five major clusters:
samples from the surface layers, and samples at 200 m and 750 m within the same site were clustered
together, respectively, while the two DCM samples formed a separate cluster. However, the DCM sample



from SS1 exhibited high similarity with the surface sample while the DCM sample from SEATS shared high
similarity with the deep layer samples from SEATS **(Fig. 2E, G).**
Regarding the horizontal distribution of bacterial community, *Cyanobacteria* was predominant in the surface
water of the SCS, accounting for 40.1, 56.2, 53.6, 41.7 and 49.6% of the total bacterial community in
SEATS, SS1, A2, B1 and C1, respectively **(Fig. 3A)**, whereas the sequences originating from
*Prochlorococcus* formed the major cluster **(Fig. 4)**. *Proteobacteria* was the second most prevalent group,
averagely accounting for 34.6% of the surface bacterial community. *Alphaproteobacteria* showed the highest
abundance (18.1% to 26.2%), while *Gammaproteobacteria* showed the second highest abundance (7.5% to
11.9%), although significantly lower than that of *Alphaproteobacteria* **(Fig. 3E)**. The families of
*Rhodospirillaceae*, *SAR86*, *OM1* clade and *SAR11* were frequently detected in the surface samples with
relative abundances ranging from 4% to 10% **(Fig. 4)**.
The vertical profiles of bacterial composition of SEATS and SS1 were then investigated **(Table S3)**.
*Proteobacteria* was the major phylum at deep layers of SEATS, accounting for 66% to 88% of the bacterial
community in the 68 m, 200 m and 750 m layers, while *Gammaproteobacteria* was the most abundant class
**(Fig. 3B, E)**. At the SS1 site, *Cyanobacteria* was relatively abundant (47.3%) in the DCM layer, whereas
*Alphaproteobacteria* represented the second largest group. At the 200 m and 750 m layers of SS1 site, the
relative abundances of the phyla *Proteobacteria, Chloroflexi* and *Marinimicrobia* increased, while
*Proteobacteria* dominated the bacterial communities. *Alpha-, Gamma-* and *Deltaproteobacteria,* which
accounted for similar proportions, were determined as the major lineages **(Fig. 3B, E)**.
The top ten most abundant OTUs at the family levels in the 16S rRNA sequencing data exhibited regional
differences between the SEATS and SS1 sites. *Prochlorococcus* dominated the surface bacterial community,
and *Acinetobacter*, with a relative abundance between 29.3% and 61.4%, represented the most abundant
OTU throughout the water column of SEATS. However, the compositions of the abundant OTUs varied
among water layers. *Flavobacteriaceae, Vibrionaceae, Prochlorococcus* and *Shewanellaceae* represented
over 40% of the bacterial community at the DCM layer of SEATS, but accounted for much smaller
percentages in the deep layers. Abundances of *Pseudoalter, Halomonadaceae* and *Alteromonadaceae* were
found to increase in the oxygen-deficient layer (750 m) of SEATS. *Prochlorococcus* (38.9%) remained as
the dominant group in the DCM layer of SS1 site. It is noteworthy that the top five most abundant OTUs in



the 200 m and 750 m layers of SS1 site were identical. The five OTUs, which accounted for 6.0% to 18.9%
of the communities, were *SAR324, SAR406, SAR202, SAR11* and *Rhodospirillaceae,* **(Fig. 4)**.
The top ten most abundant depth-dependent OTUs from SEATS and SS1 sites are listed in **Fig. 5**.
Cyanobacterial *Prochlorococcus* and *Synechococcus*, and *OM1* clade were mainly present in the euphotic
layers, whereas *Alteromonadaceae, Halomonadaceae* and *Pseudoalteromonadaceae* existed mainly in the
OMZ of SEATS. By aligning the OTUs across all depths of the SS1 site, *Oceanospirillales*, *Salinispha,*
*SAR202, SAR324* and *SAR406* exhibited prominent depth specificity in the euphotic bottom layer and OMZ.

**3.3 Diazotrophic community diversity and overall nifH composition**
A total of 131, 569 qualified reads were retrieved from ten nifH samples, and were clustered into 749
different OTUs using a sequence cutoff value of 1%. On average, the length per read was 418 bp and the
unique OTU number per sample was 181 **(Table S4)**. Surface samples, except for the samples from the
SEATS site, presented significantly higher community richness than the deep water samples, and the DCM
samples displayed the lowest community richness **(Fig. 2B)**. The within-habitat diversity of diazotrophic
communities was highest in the euphotic bottom layer of SEATS, and was also at high levels in the surface
samples of A2, B1 and C1 **(Fig. 2D; Table S4)**. Samples from the deep layers (68 m to 750 m) of SEATS
and the deep layers (105 m and 200 m) of SS1 were independently clustered and clearly separated from the
surface samples **(Fig. 2F, H)**. Notably, the surface samples shared high similarity with the deep water
samples from the SS1 site.
In contrast with the bacterial community, *Proteobacteria* was the principal phylum in all samples (40.2% on
average) except for the samples from the euphotic bottom layer (200 m) at the SEATS site, where
*Proteobacteria* only accounted for 5.5% **(Fig. 3C, D; Table S5)**. *Alphaproteobacteria* was the most
abundant group in the surface and DCM layers of the SEATS site. *Betaproteobacteria* was more prevalent in
the DCM layer (105 m) and the euphotic bottom layer (200 m) at the SS1 site, while its abundance was very
low in the samples from SEATS. While *Gammaproteobacteria* represented the most abundant class in the
OMZ of SEATS, it was detected in the surface with markedly lower abundance **(Fig. 3F)**.
*Cyanobacteria* was the dominant phylum in the DCM layer (68 m) of SEATS, accounting for 41.3% of the
diazotrophic community. *Euryarchaeota* and *Actinobacteria* were more abundant at the euphotic zone of



SEATS **(Fig. 3D)**. *Rhodobacteraceae* represented the most abundant OTU at the surface, while

*Rhodocyclales* and *Neisseriaceae* were the second abundant OTUs at the surfaces of SEATS and SS1,

respectively. Notably, the family of *Neisseriaceae* increased in relative abundance, accounting for more than

20% of the entire diazotrophic community in the DCM and euphotic bottom layers from the SS1 site.

*Chroococcales* was the dominant family in the DCM layer of SEATS, while the families of

*Rhodobacteraceae, Synechococcaceae, Rhodospirillaceae* and *Comamonadaceae* contributed substantially

to the diazotrophic composition with the portions ranging from 5% to 11%. No major N-fixing

microorganisms were found in the euphotic bottom layer of the SEATS site, but a subset of *Euryarchaeota*

groups emerged. In the 750 m layer, where oxygen was deficient, *Pseudomonadaceae* exhibited remarkable

increases in relative abundance and became the major diazotroph **(Fig. 6; Table S5)**.

Depth specificity of OTUs in the diazotrophic community was identified at each layer in the water column at

the SEATS and SS1 sites. *Desulfovibrionaceae, Cellvibrionaceae, Chromatiaceae* and *Xanthomonadaceae*

were only detected in the surface, while *Rhodospirillaceae, Rhodospirillales* and *Flavobacteriaceae* were

only observed in the DCM layer. *Actinobacteria* and *Euryarchaeota* were mainly distributed in the euphotic

bottom layer, whereas *Ardenticatenaceae, Burkholderiales* and *Pseudomonadaceae* exhibited prominent

depth specificity in the OMZ of the SEATS site **(Fig. 5C, D)**.

**3.4 Abundances and expressions of N-utilization genes in bacterial community**

Based on comparison with the EggNOG database, four groups of N-utilizing COGs in the samples were

predicted as "Transporter", "Inorganic N metabolism", "Urea metabolism" and "N fixation" **(Fig. 7; Table

S6)**. The COGs assigned to ammonium transporters (*AmtB*, COG0004), ammonium assimilation enzymes

(*GlnA*, COG0174 and *GltB*, COG0067) and amino acid transporters (*AMT*, COG0004) were the most

abundant in the surface, while abundances of COGs assigned to ureases (*UreABCDFG*,

COG0829-COG0832, COG0804, COG0378) and *nifH* (COG1348) were the second most abundant genes in

the surface. Insignificant variations and the lowest abundances of nitrate reductase and nitrite reductase

(*NirBD*, COG1251 and COG2146) were observed in different surface samples **(Fig. 7A)**.

The abundances of N-utilizing COGs varied significantly in different water layers of SEATS and SS1 **(Fig.

7B)**. The abundances of COGs attributed to *AMT, AAT, NirBD,* as well as the two ammonium assimilation





enzymes *GlnA* and *GltB*, were relatively high through the water column of SS1, especially in the surface and
DCM layers. *NifH* and *UreABCDFG* were mainly distributed in the surface and DCM layers of SS1.
Although the abundance of nitrate reductases was extremely low throughout the water column, the
abundances of subunits *NarI* (COG2181) and *NrfA* (COG3303) increased slightly in the 200 m and 750 m
layers of SS1. In addition, a large number of unclassified ABC transporters were predicted in each sample.
The relative expression levels of *amt1* and *urtA* were the highest at the surface of SEATS **(Fig. 8)**. The
expression levels of *AAT* and *GlnA* showed insignificant changes in different surface samples, except for the
lowest expression level of *AAT* in A2. High expression levels of *IdiA, ureC* and *GltS* were observed in the
surface layer of A2. In particular, the expression level of *nirA* gene, which is responsible for the assimilation
of nitrite in *Prochlorococcus,* was highest at the surface of SS1. High expression levels of *amt1, urtA* and
*AAT* were observed at the surface of SEATS. Notably, the relative abundance of *urtA* was approximately
ten-fold higher than that of *amt1* and *AAT* **(Fig.   8)**.

**3.5 Environmental influence on community diversity and structure**
The results of Mantel test and Spearman analysis showed that the temperature, salinity, dissolved oxygen
and nutrients exhibited significant correlations with the structures (Bray-Curtis distance) and
between-habitat diversities of both bacterial and diazotrophic communities **(Table 2; Table S7)**. Nutrients
were found to exert greater impacts on bacterial communities, as the concentrations of nitrate and phosphate
exhibited significant positive correlations with the within-habitat bacterial richness.

**4 Discussion**
**4.1 Spatial distribution of bacteria in the SCS basin**
In this study, high similarity in the bacterial composition among surface samples was observed, as indicated
by both the PCoA plot and UPGMA dendrogram. *Cyanobacteria* dominated the five surface samples,
comprising primarily oligotrophic *Prochlorococcus* representatives. It is well recognized that
*Prochlorococcus* is prevalently distributed in the oligotrophic oceans globally, and the distribution pattern
may contribute to its low nutrient adaptability (Jing and Liu, 2012; Garcia-Fernandez et al., 2004;
Zwirglmaier et al., 2007; Liu et al., 2007). Cai et al. (2007) and Liu et al. (2007) report the seasonal

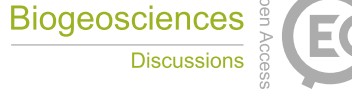

distributions of picoplankton in the nSCS, and show that the abundance of *Prochlorococcus* is higher in summer than in winter, while Xie et al. (2018) demonstrate that *Prochlorococcus* dominates the phytoplankton community in the SCS basin.

*Alphaproteobacteria* was more abundant in all surface samples while *Gammaproteobacteria* was more common in the deep layers of SEATS, in agreement with the previous studies based on 16S rRNA sequencing and metagenomic approaches (Jing et al., 2013; Tseng et al., 2015). Furthermore, *Alpha-, Gamma-* and *Deltaproteobacteria* contributed equally to the deep layer samples of SS1. These results were consistent with the clustering results, indicating that the bacterial communities in upper waters were separated from their deep-water counterpart, but the deeper-water community also exhibited geographical specificities.

Pronounced stratification among specific bacterial groups from the SEATS and SS1 sites was observed, in accordance with previous phylogenetic surveys (DeLong, 2005; Hewson et al., 2006; Treusch et al., 2009; Galand et al. 2010; Kirchman et al., 2010; Agogue et al., 2011). For instance, *SAR324, SAR406* and *SAR202* clades were relatively abundant in the euphotic and oxygen-deficient layers of SEATS and SS1. These groups are also reported as typical deep-water clades in the deep Atlantic and Pacific oceans (Wright et al., 1997; Morris et al., 2004; DeLong et al., 2006; Pham et al., 2008; Agogue et al., 2011). *SAR324*-like reads are abundant in the OMZs of the coastal regions in Iquique (Ganesh et al., 2014). Meanwhile, differences in bacterial distributions between SEATS and SS1 were observed. The family *Moraxellaceae*, comprising primarily the genus of *Acinetobacter*, was the most abundant family in the deep layers of SEATS, but was rarely detected in SS1. Jing et al. (2013) report its prevalence in SEATS, although only in the deep-water layer of 2000 m, and Xia et al. (2015) find that *Moraxellaceae* is abundant in the surface water of the SCS, and even higher *Moraxellaceae* abundance was detected in summer. In addition, *Moraxellaceae* also shows high relative abundance in the estuary ecosystem of Zhuhai (Li et al., 2018) and in the sediments of Okinawa Island (Soliman et al., 2017). The family *Moraxellaceae,* commonly found in naturally saline environments, can proliferate under a broad range of temperatures and can remineralize organic matters *in situ* (King et al. 1997; Okabe et al. 2003;Teixieraand Merquior, 2014). Nevertheless, little is known about its ecological roles, such as its role in the degradation of organic compounds. Meanwhile, few bacterial groups exhibited preferential distribution at the bottom of the euphotic layer of SEATS, characterized by



co-limitation of iron and light (Mitchell et al., 1991; Nelson and Smith, 1991). However, the groups of
*Alteronmonadaceae, Halomonadaceae* and *Pseudoalteromonadaceae* featured depth-specific distributions in
the OMZ of SEATS, while were negligible in SS1 as well as other typical OMZ bacterial communities
(Ganesh et al., 2014; Hawley et al., 2014). This discrepancy might be attributed to the perturbations brought
by Kuroshio intrusion and mesoscale eddies experienced in SEATS, since heterotrophic bacteria, particularly
*Oceanospirillales* and *Alteromonadales*, display high abundances in the Kuroshio Current and affected areas
of cyclonic eddy (Li et al. 2017; Li et al. 2018).

**4.2 Diazotrophic distribution in the SCS basin**
$N_2$ fixation provides over 10% of the total carbon production in the SCS (Voss et al., 2006). *Trichodesmium*
is considered as the main diazotroph in pelagic oceans (Moisander et al., 2008), but recent studies showed
that non-cyanobacterial diazotrophs and unicellular cyanobacteria groups are also present and active (Zhang
et al., 2011; Moisander et al., 2014; Li et al., 2018). A prominent feature of the SCS during our sampling
period was that diazotrophic *Alphaproteobacteria*, comprising primarily *Rhodobacteraceae*, dominated the
SCS surface while the abundance of the unicellular *Cyanobacteria* was negligible, in agreement with Zhang
et al. (2010), who report the dominance of *Alphaproteobacteria* and lower abundance of both
*Trichodesmium* and heterocystous cyanobacterial diatom symbionts in the SCS deep basin area.
Inconsistent with the findings of Zhang et al. (2010), Moisander et al. (2008) and Moisander et al. (2014),
*Gammaproteobacteria* was rare in our samples except for the OMZ sample of SEATS. Instead,
*Betaproteobacteria* was widely distributed at the SEATS and SS1 sites across all depths. In particular,
*Betaproteobacteria*, comprising primarily *Neisseriaceae*, overwhelmingly dominated the diazotrophic
communities in the DCM and euphotic bottom layers of SS1. *Betaproteobacteria* is a typical freshwater
lineage (Zwart et al., 2002) frequently present in the oceans (Brown et al., 2009; Andersson et al., 2010).
Phylogenetic analysis of our study revealed that the proportion of *Betaproteobacteria* in the SCS was much
smaller, indicating that the survey area was not affected by offshore freshwater inflows. Nevertheless, a
concern is that nifH-like sequences related to *Betaproteobacteria* may be present in the PCR reagents (Zehr
et al., 2003; Goto et al., 2005). Therefore, the identification of *Betaproteobacteria*-like sequences should be
treated with caution.



In this study, the stratified distribution of bacterial assemblages in SEATS differed from that in SS1.
*Chroococcales* dominated the diazotrophic community of the DCM layer at the SS1 site. This group is also a
major diazotroph in Arctic and Indian seawaters (Diez et al., 2012; Bauer et al., 2007). $N_2$-fixing archaea is
confined only to the phylum *Euryarchaeota* (Dos-Santo et al., 2012). The group II *Euryarchaeota* (*MG-II*)
emerged at the bottom of the euphotic layer of SEATS, accounting for more than 10% of the bacterial
community. *MG-II* is distributed within the euphotic zone of temperate waters and plays pivotal roles in
marine N cycling (Haro-Moreno et al., 2017; Qin et al., 2014). The presence of the *MG-II* groups inferred
the existence of an *Euryarchaeota*-leading diazotrophic community at the bottom of the euphotic zone of
SEATS, and confirmed the ecological significance of marine archaea as a new N contributor in deep oceans.
The bottom of the euphotic zone is characterized by specific dissolved organic matters, and further
investigation is needed to evaluate it as a potential habitat for *MG-II*.

**4.3 Environmental influences on bacterial community**

The gradient physico-chemical characteristics of the water column, such as the declines in light intensity and
temperature, as well as the scarce organic matter availability, have been identified as crucial factors
impacting the vertical distribution of bacterial communities (Giovannoni et al., 2005; DeLong et al., 2006).
The depth and latitude also represent highly significant explanatory variables for the bacterial populations
from different water masses in the North Atlantic Ocean (Agogue et al., 2011). In the present study,
remarkable differences in bacterial and diazotrophic compositions were identified for SEATS and SS1
located at the northern and southern basin regions of the SCS. Temperature, salinity, dissolved oxygen and
nutrient concentrations contributed in synergy to the horizontal and vertical variations of bacterial structures,
in agreement with the findings of previous studies that the physico-chemical parameters lead to almost
identical results, as the vertical stratification in the Northwestern Mediterranean Sea (Ghiglione et al., 2008)
and variations of major phytoplankton groups in the SCS are influenced by temperature, irradiance and
nutrient concentrations (Zhang et al., 2014; Xiao et al., 2018). In our study, the bacterial community in the
DCM layer of SS1 was highly similar to that of the surface water, likely due to the strong vertical mixing in
SS1 induced by the tropical storm Merbok that passed through the sampling area.
Both bacterial and diazotrophic communities shared similarities in the deep layers, but were distinct in the





upper layers, suggesting that the deep-sea assemblages formed a separate cluster from the surface
assemblages as revealed in the North Pacific Ocean (Brown et al., 2009). As indicated by the results of the
correlation analysis, the concentrations of nitrate and phosphate were the key factors affecting the richness
and diversity of bacterial community. Depletion of nutrients, particularly phosphate, in the upper waters
contributed to the low richness and diversity of bacterial communities.

**427  4.4 N use strategies of bacterial community in the SCS**

PICRUSt prediction and real-time qPCR are valuable tools to assess gene expressions in microorganisms
from the natural environments (Langille et al., 2013; Li et al., 2018). In epipelagic waters of the SCS, N is
one of the limiting nutrients, in contrast to the dark, energy-limited but relatively N-rich deep oceans (Batut
et al., 2014; Giovannoni and Nemergut, 2014). The dominant group, *Prochlorococcus*, plays critical roles in
marine N cycle of the SCS. To investigate the expressions of N-utilization-related genes in the bacterial
communities and *Prochlorococcus* in different regions of the SCS basin, we examined both the gene
abundances and expression levels of transporters and N utilization pathways for both the inorganic and
organic N sources. Although *IdiA* is not directly involved in the N utilization, the expression of *IdiA* was
also taken into account in our study, as iron is essential for nitrite assimilation and N fixation in the cells.
The results of PICRUSt prediction and qPCR analysis revealed that the amino acid transporters and
ammonium assimilating enzymes were prevalent in the surface of SCS, indicating that the amino acids
represent a major N source for the bacterial community, consistent with the results of Zubkov et al. (2003)
and García-Fernández et al. (2004). As reported previously, multiple protein biomarkers from
*Prochlorococcus* provide indications of nutritional stress, for instance, the urea transporter for nitrogen and
IdiA for iron (Saito et al., 2014). The expression of urease complex is also up-regulated under N deprivation
(Tolonen et al., 2006). Although urea transporter expression was not predicted by PICRUSt, highest
expression levels of ureases and ammonium transporters were detected in the surface waters. Consistently,
expression levels of urea transporters and ammonium transporters were also relatively high in the surface
layer of SEATS, while *IdiA* and urease exhibited the highest expression levels in A2, suggesting N
deficiency in SEATS and deficiencies of both iron and N in A2. Ammonium, urea and amino acids were the
major N sources for the bacterial community in SEATS, while urea was the major N source in A2. Since iron




is an indispensable metal coenzyme for nitrate/nitrite reductase and nitrogenase, it is speculated that nitrate/nitrite reduction and $N_2$ fixation are limited by iron deficiency, which in turn promotes urea utilization in the cells. In the equatorial Pacific, both urea transporters and IdiA from *Prochlorococcus* are among the most abundant proteins, while the urease and urea transporter operons are present at high abundances in the *Prochlorococcus* clades from iron-depleted oceanic regions of the Eastern Equatorial Pacific and Indian Ocean, implying that the dissolved organic N is an important nutritional source for *Prochlorococcus* in the iron-limited regions (Rusch et al., 2010; Saito et al., 2014).

Although most phytoplankton species can use ammonia, nitrite and nitrate as sole nitrogen sources, nearly all *Prochlorococcus* isolates use ammonia as their N source except for two low light-adapted *Prochlorococcus* clades (eNATL and eMIT9313), which can also assimilate nitrite (Moore et al., 2002; Martiny et al., 2009). The availability of nitrite may therefore influence the distribution of these two clades, although relevant evidences have not been reported in the field (Bouman et al., 2006; Johnson et al., 2006). In our study, the abundances of *nitrate/nitrite reductases* were extremely low throughout the entire water column of the SCS, even in the nitrate-rich deeper waters, indicating that $NO_X$-N was not the main N source for the bacterial communities of the SCS basin. However, the qPCR results revealed that the expression level of nitrite reductase was exclusively high in the surface layer of SS1, indicating that a different N utilization strategy with an emphasis on nitrite might exist within SS1 and other sites.

## 4.5 Conclusion and recommendations

Similar horizontal distribution patterns of both bacterial and diazotrophic compositions were observed in the surface of the SCS basin, while different N utilization strategies were found to exist in the bacterial communities and *Prochlorococcus*. Moreover, the bacterial communities and N utilization strategies varied among the typical water masses under the influences of physical and hydrochemical conditions along the water column. Meanwhile, different prevalent OTUs were identified at different depths among the regions of nSCS and sSCS, under the influence of the Kuroshio intrusion into the nSCS basin. The depth was found to be a highly significant explanatory variable for the bacterial populations from different water masses. Given the high spatial heterogeneity inherent to marine environments and the consequent variations of bacterial community structures, a comprehensive study, such as metagenomics and metaproteomics, not only could



provide exhaustive characterization of bacterial assemblages, but also would aid the identification of specific
bacterial groups and metabolic pathways, thus revealing the specific ecological roles of marine bacterial
communities (Venter et al., 2004; DeLong et al., 2006).

*Data availability*. Data are available in GenBank under BioProjectID PRJNA509084. The individual accession
numbers SAMN10537119-10537129 represented 16S rRNA libraries and SAMN10537130-10537139 represented
nifH libraries.

*Author contributions*. WDZ and LYY conceived and designed this study. LYY, WPF and LDX conducted the field
work. LYY and CXH analyzed the data. LL contributed to the instrumental analysis. LYY and CXH drafted the paper,
and WDZ revised and finalized the paper.

*Competing interests*. The authors declare that they have no conflict of interest.

*Acknowledgements*.
This study was supported by the National Natural Science Foundation of China through grant 41425021, and
the Ministry of Science and Technology through grant 2015CB954003. D.-Z. Wang was also supported by
the 'Ten Thousand Talents Program' for leading talents in science and technological innovation.

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



**Tables**

**Table 1** Sampling sites and physicochemical parameters.

| Stations | Depth | Latitude (°N) | Longitude (°E) | Temperature (°C) | Salinity (PSU) | Chl a (mg/m³) | DO | $NO_2^-$ | $NO_3^-$ | $PO_4^{3-}$ | $SiO_3^{2-}$ |
|---|---|---|---|---|---|---|---|---|---|---|---|
| SEATS | 4m | 18.00 | 116.00 | 29.81 | 33.69 | 0.116 | 5.79 | BTD | BTD | BTD | 1.67 |
| | 68m | | | 22.57 | 34.57 | 0.899 | 6.65 | BTD | 0.25 | BTD | 2.74 |
| | 200m | | | 14.18 | 34.53 | 0.004 | 4.8 | BTD | 17.56 | 1.20 | 25.77 |
| | 750m | | | 5.90 | 34.47 | 0.026 | 2.75 | BTD | 35.98 | 2.62 | 104.33 |
| SS1 | 4m | 14.00 | 116.00 | 30.23 | 33.34 | 0.158 | 6.13 | BTD | BTD | BTD | 2.00 |
| | 105m | | | 22.29 | 34.53 | 0.623 | 5.28 | 0.05 | 3.17 | 0.42 | 6.82 |
| | 200m | | | 14.85 | 34.54 | 0.013 | 4.052 | BTD | 16.45 | 1.08 | 20.21 |
| | 750m | | | 6.10 | 34.46 | 0.044 | 2.57 | BTD | 34.78 | 2.62 | 105.32 |
| A2 | 6m | 12.00 | 116.00 | 30.13 | 33.44 | 0.258 | 6.06 | BTD | BTD | BTD | 1.96 |
| B1 | 6m | 14.00 | 113.00 | 29.97 | 33.54 | 0.084 | 6.13 | BTD | BTD | BTD | 2.10 |
| C1 | 4m | 12.00 | 113.00 | 29.96 | 33.40 | 0.139 | 6.09 | BTD | BTD | BTD | 2.13 |

PSU, practical salinity unit; Chla, chlorophyll a. "BTD" means the value is below the detection limit. The upper measuring limits of the AA3 Analyzer as referred to $NO_2^-$, $NO_3^-$, $PO_4^{3-}$ and $SiO_3^{2-}$ are 0.04 μM, 0.1 μM, 0.08 μM and 0.16 μM, respectively.





**Table 2** The spearman's correlations between bacterial and diazotrophic community, and environmental factors.

| | | | | | Spearman's correlation | | | |
|---|---|---|---|---|---|---|---|---|
| | | T | Salinity | Chl a. | $NO_2^-$ | $NO_3^-$ | $PO_4^{3-}$ | $SiO_3^{2-}$ | DO |
| Bacterial diversity | OTUs | -0.518 | 0.200 | -0.700* | -0.100 | 0.648* | 0.778** | 0.591 | -0.793** |
| | Ace | -0.445 | 0.077 | -0.773** | -0.300 | 0.563 | 0.689* | 0.573 | -0.692* |
| | Shannon index | -0.327 | 0.465 | -0.273 | 0.100 | 0.181 | 0.243 | 0.200 | -0.323 |
| | PCoA axis 1 | -0.0909** | 0.761** | -0.600 | 0.000 | 0.915** | 0.843** | 0.800** | -0.692* |
| Bacterial structure | Bray-Curtis distance | 0.7974** | 0.2829* | 0.092 | -0.154 | 0.7653** | 0.700** | 0.737** | 0.673** |
| Diazotrophic diversity | OTUs | 0.394 | -0.565 | -0.333 | -0.522 | -0.200 | -0.089 | -0.079 | 0.225 |
| | Ace | 0.697* | -0.863** | -0.176 | -0.522 | -0.627 | -0.464 | -0.491 | 0.413 |
| | Shannon index | -0.079 | -0.316 | -0.285 | -0.290 | 0.162 | 0.225 | 0.333 | -0.061 |
| | PCoA axis 1 | -0.685* | 0.766** | -0.127 | 0.406 | 0.769** | 0.676* | 0.648* | -0.529 |
| Diazotrophic structure | Bray-Curtis distance | 0.564* | 0.227 | 0.219 | 0.265 | 0.545* | 0.519* | 0.542* | 0.505* |

**P < 0.01 and *P < 0.05 indicate significant correlation.



**Figure legends**

**Figure 1.** Sampling sites of the South China Sea basin

**Figure 2.** Differences in bacterial and diazotropic community richness, diversity and structure from horizontal and vertical bacterial samples in the Northern and Southern SCS: (a, b) The richness of the bacterial community and the diazotropic community; (c, d) The diversity of the bacterial community and the diazotropic community; (e, f) Principal Coordinate Analysis (PCoA) of the bacterial community and the diazotropic community; (g, h) Hierarchical clustering tree on 16s rRNA OTU level and on nifH OTU level.

**Figure 3.** Relative abundances of bacterial and diazotropic compositions of the nSCS and sSCS basin at phylum level: (a-b) Horizontal and vertical bacterial composition; (c-d) Horizontal and vertical diazotropic composition; (e-f) Taxonomic groups of *Proteobacteria* in bacterial and diazotropic community. *unclassified.

**Figure 4.** Relative abundances of the 10 most abundant OTUs at the family level in the horizontal and vertical bacterial samples of the nSCS and sSCS basin. *unclassified. *Gammaproteo* represents *Gammaproteobacteria*; *Pseudoalter* represents *Pseudoalteromonadaceae*; *Sphingomon* represents *Sphingomonadaceae*; *Salinispha* represents *Salinisphaeraceae*.

**Figure 5.** Top ten most abundant depth-specific OTU groups of the bacterial (a, b) and diazotrophic communities (c, d) from vertical nifH samples in the nSCS and sSCS basin. The area of each bubble represents the cumulative relative abundance in the sample examined; *unclassified.

**Figure 6.** Relative abundance of the 10 most abundant OTUs at the family level in the horizontal and vertical nifH samples of the nSCS and sSCS basin. *unclassified.

**Figure 7.** Horizontal (a) and vertical (b) distributions of N utilization genes predicted according to the bacterial OTUs in the nSCS and sSCS basin.

**Figure 8.** Relative transcripts of N utilization genes and relative transcripts of *amt1, urtA* and *AAT* in *Prochlorococcus* among different surface samples. Error bars represent the standard deviations of the values generated from three biological repeats.



Figure 1





Figure 2



(a)

(b)

(c)

(d)

(e)

(f)

(g)

(h)





Figure 3




Figure 4





Figure 5



Figure 6




Figure 7

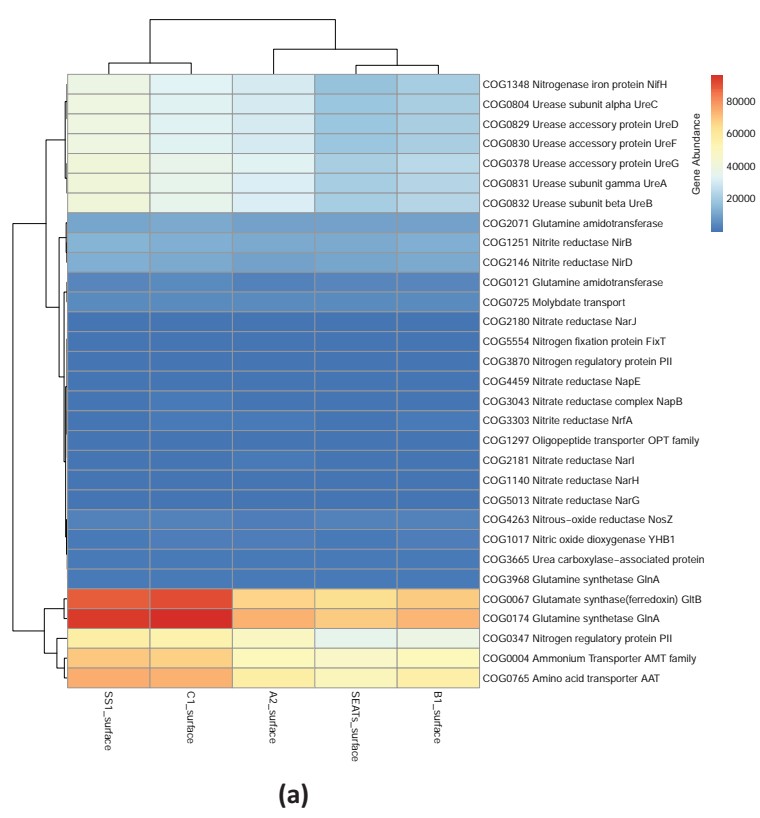

**(a)**

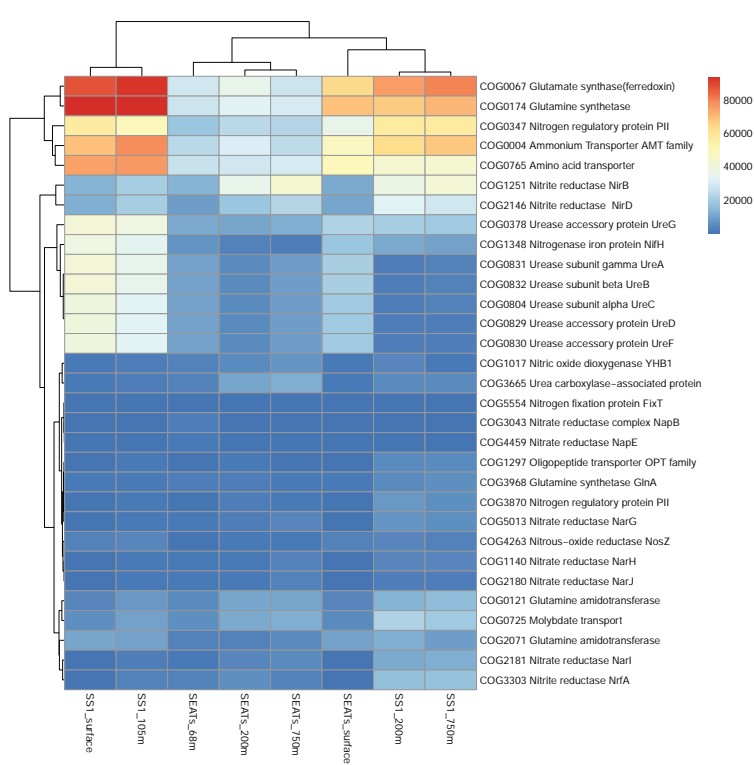

**(b)**





Figure 8

