# Peer review of "Diversity, distribution and nitrogen use strategies of bacteria in the South China Sea basin"

_Biogeosciences, 2018_

## Referee Comment (RC1) · Anonymous Referee #1 · 3 Apr 2019

The manuscript presents data from amplicon sequencing of a subunit of 16S rDNA and the nifH gene, the functional molecular marker gene for dinitrogen (N2) fixation. Further, a set of quantitative molecular data is presented on genes involved in nitrogen turnover and (surprisingly firstly mentioned in detail in the discussion) of iron acquisition. A statistical analysis was applied to explore the parameters to which the microbial community responds. The paper, as presented, provides a mere description of a lot of more or less expected facts, in line with previous studies from the South China Sea (SCS). Besides the fact that references are missing for various claims in the introduction and discussion parts, a clear storyline is missing. The manuscript will need major rewriting and focusing on one topic in order to make its significance visible. I would recommend to focus on either the community dynamics in the SCS, or on nitrogen re-

generation in those waters, to apply a more thorough description of the methods and a true discussion instead of a repetition of other studies' results. In this context, I have very strong doubts on the validity of the nifH dataset because the primers used in this study have been described to have an extreme bias in Ocean environments. I am not convinced that this dataset is representative. Similarly, the methods for the 16S rDNA amplicon analysis make no sense the way they are described. This needs substantial clarification. The conclusion does not provide additional insights, so also here re-writing is necessary. In addition to this, some aspects of the writing come across sloppy, including gene names, which are commonly written in italics, the use of the term N fixation, while N2 fixation is meant, or the awkward differentiation between bacteria and diazotrophs. Diazotrophs are mostly bacteria, besides bacteria only methanogenic archaea possess the genes for N2 fixation. A language editing is needed.

Below I have some more specific suggestions for the different parts of the manuscript:

Abstract

The expression 'layers' may appear intuitive, however, it took me a while to figure out if we are talking about the water column or sediment.

There is no line of reasoning or storyline visible, just a collection of facts. The last sentence is out of place and it is unclear on what this claim is based.

Introduction

l. 59 sentence says nothing

l. 61: not only for bacteria, but for life in general. Also, the sentence sounds awkward. References are missing

l. 64 I don't understand the sentence

l. 66 N2 fixation is not an adaptation but an evolutionary development

l. 73 How does this show up?

l. 73 What do you mean by metagenomics, here?

l. 74 revealed; generally please check the use of the correct tenses.

l. 87 coastal

l. 88 failed; key or new- what do you mean with this? Using metagenomics without single cell rates there is no way to do this. Also references are missing here.

l. 101 PICRUSt- please explain

Methods l. 111 Why do you use was? Has it been moved?

l. 112, 113 seawater from. . .

l. 114 was instead of were

l. 116 Do you want to say you collected the cells on the 0.2um filter?

l. 117 You stored on board until analysis?

l. 119 Which sensor did you use?

l. 121 an instead of the

l. 122 Germany was almost certainly not the company you bought the analyzer from, please give the correct name, what does this reference indicate here, this doesn't seem to be correct.

l. 124 this is a GF/F filter, and not a membrane. Please also provide the extraction method and the calibration details, including the type of standard you used.

l. 127 So you are saying you put the filters directly in the tubes in the Fast DNA spin kit? This can't work. Please describe how you extracted from the filters.

l. 128 replicates, not repeats

l. 129 reagent instead of regent; please give details on the Trizol-chloroform extraction.

This is insufficient to reproduce it. How was the leftover DNA removed, and how was the purity checked?

l. 138 f Here, we are running into a problem. Gaby et al. 2012 showed that this primer set is covering only 5% of nifH diversity in the sea. So, this makes this part of the dataset actually non-credible. You are coming back to this in l 382, where you show inconsistencies with other datasets, and you are discussing the potential of contamination- this could have been outruled by non-template controls. If you did them please provide the results. If not the dataset has almost no credibility.

Gaby JC, Buckley DH (2012) A Comprehensive Evaluation of PCR Primers to Amplify the nifH Gene of Nitrogenase. PLoS ONE 7(7): e42149. https://doi.org/10.1371/journal.pone.0042149

l. 144 replace the by a, provide details on the kit and protocol for the library construction, plus on the length of the reads

l. 146 With which program, how?

l. 157 ff: Gene names start with small letters

l. 159 Provide a reference, details on the quality checks. What do you mean with control genes? Is this to quantify against? If so I am not convinced of the quantification either. A quantification of a Prochlorococcs gene against a general cyanobacterial 16S amplicon does not make sense to me.

l. 170 remove 'levels of'

l. 172 What do you mean by holes?

l. 181 this needs a reference

l. 191 ff How exactly do you do a functional prediction based on 16S data, and somehow on the metagenome? Then again you use the 16S-only Greengenes database. Then you transfer to EggNOG, which is a functional annotation pipeline- what did you

transfer. This is completely unclear to me. Also, it is necessary to put references here. Why did you not just Kegg-map directly from the metagenomes? This, to me, would be more credible.

l. 197 Which bioinformatics analysis?

l. 200 Hydrochemical?

l. 201 What is an upper mixed layer, could that be substantiated ?

l. 212 What was the aim of this characterization?

l. 213 I would like to see a table with reads per sample.

l. 215 How do you know about the coverage?

l. 227 and throughout the results If you use bacteria in plural then use were instead of was

From l. 227 to l. 259: this is a listing of numbers, that are in parts contradictory. It is a bit unclear why you are reporting on all the groups without saying much. Please streamline this.

l. 262 What are qualified reads?

l. 263 this means a 99% identity?

l. 265 How was the richness determined?

l. 272 to instead of with

l. 288 they didn't emerge, they were detected. Also it's not a subset but a clade.

l. 289 this is not remarkable but rather expected, too.

l. 304 I frankly don't believe that nifH is the second most abundant gene anywhere.

l. 322 ff what you are saying here is that all parameters you included were significantly

correlated to the community structure. I am not quite sure how this informs us.

l. 388 How to infer a proportion here?

l. 395 This is most likely because only one group of euryarchaeota is able to fix N2-however, they have never been shown to actively do so in the water column. Thus the statement in line 401 is way too speculative.

l. 424 How do you know this? The statistics show you a correlation to the selection of parameters you used in the test. This needs more of a discussion. Also it is somewhat contradicting your conclusion

---

## Referee Comment (RC2) · Anonymous Referee #2 · 8 Apr 2019

This study focuses on the 16S and nifH community structure of the South China Sea. It is a description study with a few sampling points at various depths. The main problem of this paper, is the lack of a narrative. The reader will not be sure what is the main finding of this study against already acquired knowledge. The nifH amplicon approach could be mentioned, but as the nifH primers can only capture a small proportion of the community, the findings could be used in support of other results, rather than forming the main findings narrative.

In general, the study is worth publishing as it describes the community snapshot of the SCS. However, this paper needs re-writing in order to better show the importance of the findings.

Introduction: Could you add a few a paragraph about how does the SCS microbiome

structure, N-fixation and nutrient limitation etc. compares with other seas. As a reader I would like to know either SCS is an anomaly or does it represent a typical coastal microbiome.

Results: I find the use of percentage values almost irrelevant considering that you have a single biological replicate per sampling site/depth. I can agree that there is a pattern of depth vs surface vs location at the global community level (the whole community pattern as represented on PCoA), but the percentage differences would definetely change with more replications. Please, add that the single replicate does not allow for any statistical analysis to be conducted and instead of using numbers, please just state either some phyla/genera seem to be more/less abundant for a specific sampling point.

Results: I would like to see a PCA or similar analysis linkning nutrient, salinity, temperature with the community structure. Is the influence of the nutrients, salinity etc. smaller or greater than the location? Can you separate them? What I am asking, is the knowledge about salinity and nutrients status of the sampling location enough to predict the likely microbial community structure?

(a small remark) please use 100,000 instead of 100, 000 in your sequencing number reports

For the nifH part, please clearly state that no nifH primers are able to provide a comprehensive nifH community profile. Different studies chose different primers. You are unravelling a part of nifH community. While the comparisons between sites are valid, please remember that this is just a part of the community, and quite likely most of this community is still our of our reach.

Figure 1, please add a legend, what depth does the colour signify

Figure 2a-d please increase the font size of the labels (graph bottoms)

Figure 3cd plase correct *Bacteira to Bacteria, please explain, which groups are included in this category in the figure legend.

---

## Author Comment (AC1) · 28 Apr 2019

Reviewer 1 comments: The manuscript presents data from amplicon sequencing of a subunit of 16S rDNA and the nifH gene, the functional molecular marker gene for dinitrogen (N2) fixation. Further, a set of quantitative molecular data is presented on genes involved in nitrogen turnover and (surprisingly firstly mentioned in detail in the discussion) of iron acquisition. A statistical analysis was applied to explore the parameters to which the microbial community responds. The paper, as presented, provides a mere description of a lot of more or less expected facts, in line with previous studies from the South China Sea (SCS). Besides the fact that references are missing for various claims in the introduction and discussion parts, a clear storyline is missing. The manuscript will need major rewriting and focusing on one topic in order to make its significance

visible. I would recommend to focus on either the community dynamics in the SCS, or on nitrogen re-generation in those waters, to apply a more thorough description of the methods and a true discussion instead of a repetition of other studies' results. In this context, I have very strong doubts on the validity of the nifH dataset because the primers used in this study have been described to have an extreme bias in Ocean environments. I am not convinced that this dataset is representative. Similarly, the methods for the 16S rDNA amplicon analysis make no sense the way they are described. This needs substantial clarification. The conclusion does not provide additional insights, so also here re-writing is necessary. In addition to this, some aspects of the writing come across sloppy, including gene names, which are commonly written in italics, the use of the term N fixation, while N2 fixation is meant, or the awkward differentiation between bacteria and diazotrophs. Diazotrophs are mostly bacteria, besides bacteria only methanogenic archaea possess the genes for N2 fixation. A language editing is needed.

Answer: Thanks for the reviewer's comments and suggestions. We made extensive changes to this manuscript and re-wrote it according to the reviewer's suggestions. Firstly, the revised manuscript was focused on the community dynamics of the SCS according to the reviewer's suggestion, and both the contents related to nifH and nitrogen utilization were removed from our new manuscript. Moreover, redundancy analysis between environmental factors and community structure (Fig. 1) as well as the overall PICRUSt prediction (Fig. 2; Fig. 3) were conducted, and these results were added to the revised manuscript. Secondly, we revised the Methods section substantially, including the 16S amplicon, nucleic extraction, PICRUSt prediction and statistical analysis used in this study. We also added more discussions to the Discussion section, such as discussions about environmental influence and community functional potentials. New conclusions were re-written based on the results. Briefly, our study reported different distribution and function of bacterial community between the nSCS and the sSCS in the SCS basin. We emphasizes the importance of environmental factors on community structure and provided evidence that the SCS basin exhibited different functional zonations among depths which enriched different metabolic potentials. Thirdly, we re-constructed the storyline consisting mainly of "16S rRNA", "environmental influence" and "PICRUSt predictions" in the revised manuscript. A language editing was also done on the revised manuscript.

Abstract The expression 'layers' may appear intuitive, however, it took me a while to figure out if we are talking about the water column or sediment. There is no line of reasoning or storyline visible, just a collection of facts. The last sentence is out of place and it is unclear on what this claim is based.

Answer: Thanks for the reviewer's comments. The abstract was re-written in the revised manuscript. We summarized our main findings in a more clear way and proposed the conclusions entirely basing on the present results. Some vague expressions were removed.

Introduction l. 59 sentence says nothing Answer: Deleted in our revised manuscript.

l. 61: not only for bacteria, but for life in general. Also, the sentence sounds awkward. References are missing Answer: Deleted in our revised manuscript.

l. 64 I don't understand the sentence Answer: Deleted in our revised manuscript.

l. 66 N2 fixation is not an adaptation but an evolutionary development Answer: Deleted in our revised manuscript.

l. 73 How does this show up? Answer: Thanks for the reviewer's comment. Previous studies indicate that phytoplankton growth is inhibited by nitrogen and/or phosphorus limitation in the SCS (Wu et al., 2003; Chen et al., 2004), supplement of nitrate or phosphate results in a phytoplankton bloom through enhancing chlorophyll a concentration, primary and new productions. References were added to our revised manuscript.

l. 73 What do you mean by metagenomics, here? Answer: The original intention was to introduce the method used by that paper. It was removed from the revised manuscript.

l. 74 revealed; generally please check the use of the correct tenses. Answer: Thanks for the reviewer's comment and we corrected tenses throughout the revised manuscript.

l. 87 coastal Answer: Corrected.

l. 88 failed; key or new- what do you mean with this? Using metagenomics without single cell rates there is no way to do this. Also references are missing here. Answer: Deleted

l. 101 PICRUSt- please explain Answer: The Phylogenetic Investigation of Communities by Reconstruction of Unobserved States (PICRUSt) is an algorithm and software package that performs functional predictions for 16S sequences, using a reference phylogeny to weight the relative functional contributions of closely related sequence genomes (Douglas G. M. 2018, pp169-177). Related information was added to the revised manuscript.

Methods l. 111 Why do you use was? Has it been moved? Answer: Corrected in the revised manuscript.

l. 112, 113 seawater from: : : Answer: Corrected in the revised manuscript.

l. 114 was instead of were Answer: Corrected in the revised manuscript.

l. 116 Do you want to say you collected the cells on the 0.2um filter? Answer: Yes, bacterial cells between 0.2-3 um were collected onto the 0.2 um filter membrane and then the filter membrane was properly preserved.

l. 117 You stored on board until analysis? Answer: Yes, the filter membranes were immediately frozen in liquid nitrogen, and then stored in the -80 °C refrigerator on board until analysis.

l. 119 Which sensor did you use? Answer: The SBE 911 (Sea Bird) was equipped with these sensors, including temperature, conductivity, pressure, oxygen, light transmission and fluorescence. More details were added to the Method section.

l. 121 an instead of the Answer: Corrected.

l. 122 Germany was almost certainly not the company you bought the analyzer from, please give the correct name, what does this reference indicate here, this doesn't seem to be correct. Answer: Yes, we bought the analyzer (AutoAnalyzer 3, Bran Luebbe Gmbh) from Germany. We removed the reference in the revised manuscript.

l. 124 this is a GF/F filter, and not a membrane. Please also provide the extraction method and the calibration details, including the type of standard you used. Answer: Thanks for the reviewer's comment. The original descriptions were revised according to the reviewer's suggestion. The detailed extraction and calibration method was added. Briefly, seawater for chlorophyll a (Chl a) determination were filtered onto a 25-mm-diameter glass fiber filters. The filter was extracted using 90% acetone for Chl a analysi. After 16-24 h at -20 °C in a dark environment, the Chl a was measured using a Trilogy fluorometer (Turner Designs, USA) and calibrated using the standard curve with the Chl a standard (DHI lab, Denmark) according to the method reported by Welschmeyer (1994).

l. 127 So you are saying you put the filters directly in the tubes in the Fast DNA spin kit? This can't work. Please describe how you extracted from the filters. Answer: Thanks for the reviewer's comment. Filter of each sample was thawed on ice and cut into scraps using a sterilized scissor. Then they were lysed using the FastDNA SPIN Kit (MP Laboratories, Inc.) with beads beating in a homogenizer (06404-200-RD000, Bertin Minilys) for 5 min. Then DNA was extracted and purified following the protocol of the manufacturer. Detailed information was added to the Method section.

l. 128 replicates, not repeats Answer: Corrected in the revised manuscript.

l. 129 reagent instead of regent; please give details on the Trizol-chloroform extraction. This is insufficient to reproduce it. How was the leftover DNA removed, and

how was the purity checked? Answer: Corrected. The details of RNA extraction were deleted from the manuscript since the quantitative PCR results were removed from the manuscript. The purity of DNA in this paper was checked by the values of OD260/OD280 in a NanoVue Plus spectrophotometer.

l. 138 f Here, we are running into a problem. Gaby et al. 2012 showed that this primer set is covering only 5% of nifH diversity in the sea. So, this makes this part of the dataset actually non-credible. You are coming back to this in l 382, where you show inconsistencies with other datasets, and you are discussing the potential of contamination- this could have been outruled by non-template controls. If you did them please provide the results. If not the dataset has almost no credibility. Gaby JC, Buckley DH (2012) A Comprehensive Evaluation of PCR Primers to Amplify the nifH Gene of Nitrogenase. PLoS ONE 7(7): e42149. https://doi.org/10.1371/journal.pone.0042149 Answer: Thanks for the reviewer's comments. We agreed that the primers used here might cause the biased results. This part was removed from the revised manuscript.

l. 144 replace the by a, provide details on the kit and protocol for the library construction, plus on the length of the reads. Answer: Corrected. The purified 16S rRNA amplicons were sequenced using paired-end sequencing (2 × 300) and the MiSeq Reagent Kit v2 (500 cycle, Illumina, San Diego, CA, USA) on a Illumina MiSeq platform. Raw fastq files were quality-filtered by Trimmomatic and merged by FLASH with the following criteria: (i) The 300bp reads were truncated at any site receiving an average quality score <20 over a 50 bp sliding window. (ii) Sequences with overlap longer than 10 bp and mismatch no more than 2 bp were merged. (iii)Sequences of each sample were separated according to barcodes (exactly matching) and Primers (allowing 2 nucleotide mismatching), and reads containing ambiguous bases were removed. Operational taxonomic units (OTUs) were clustered with 99% similarity cutoff using UPARSE (v7.1 http://drive5.com/uparse/) with a novel 'greedy' algorithm that performs chimera filtering and OTU clustering simultaneously. The taxonomy of each 16S rRNA gene sequence was analyzed by RDP Classifier algorithm (http://rdp.cme.msu.edu/) against

the Silva (SSU128) rRNA database (Release 128, http://www.arb-silva.de). Detailed information was added to the Method section.

l. 146 With which program, how? Answer: The program UPASE (v7.1) was used to cluster OTUs. Detailed information was added to the Method section.

l. 157 ff: Gene names start with small letters Answer: We deleted gene names for this part was removed from the revised manuscript.

l. 159 Provide a reference, details on the quality checks. What do you mean with control genes? Is this to quantify against? If so I am not convinced of the quantification either. A quantification of a Prochlorococcs gene against a general cyanobacterial 16S amplicon does not make sense to me. Answer: Thanks for the reviewer's comments. This part was removed from the revised manuscript.

l. 170 remove 'levels of' Answer: Corrected.

l. 172 What do you mean by holes? Answer: We deleted it for this part was removed from the revised manuscript.

l. 181 this needs a reference Answer: We deleted it for this part was removed from the revised manuscript.

l.191 ff How exactly do you do a functional prediction based on 16S data, and somehow on the metagenome? Then again you use the 16S-only Greengenes database. Then you transfer to EggNOG, which is a functional annotation pipeline- what did you transfer. This is completely unclear to me. Also, it is necessary to put references here. Why did you not just Kegg-map directly from the metagenomes? This, to me, would be more credible.

Answer: Thanks for the reviewer's comments. More details and references about the PICRUSt methods were added. Briefly, the prediction is carried out in four steps. First, 16S sequences are against the Greengene database to obtain "reference OTU"; Second, 16S sequences are standardized using the R package; Third, "reference OTU"

is mapped to the EggNOG or KEGG to get functional abundances; Last, multiplying standardized OTU abundance by that OTU's functional abundance (to form the suppositional metagenome rather than sequenced metagenome). The PICRUSt is developed based on the Greengenes database, that's why we have to map the 16S sequence to the Greengenes database first. In the revised manuscript, metagenomic functional predictions were assigned to KEGG Orthology (KO) for all genes. We used a bar graph (Fig. 2) to show the distribution of KEGG-pathways and several heat-maps (Fig. 3) to show the vertical distributions of KEGG-tier 3 KOs. Related contents were added to the Result and Discussion sections.

l. 197 Which bioinformatics analysis? Answer: We revised our manuscript and removed related sentences in our new version.

l. 200 Hydrochemical? Answer: Corrected it.

l. 201 What is an upper mixed layer, could that be substantiated ? Answer: Thanks for the reviewer's comment. The upper mixed layer in the SCS is generally driven by the horizontal eddy heat flux and strong winds (Pan and Sun, 2018, doi: 10.1038/s41598-018-33803-2). During our cruise, it was substantiated through temperature and salinity of seawater and the numerical simulation of circulation.

l. 212 What was the aim of this characterization? Answer: We revised our manuscript and removed related sentences in our new version.

l. 213 I would like to see a table with reads per sample. Answer: Thanks for the reviewer's suggestion. The table containing reads per sample was included in the supplementary file. Please see Table 1.

l. 215 How do you know about the coverage? Answer: We calculated the coverage using the Mothur software (v1.30.1) (http://www.mothur.org/wiki/Schloss_SOP#Alpha_diversity).

l. 227 and throughout the results If you use bacteria in plural then use were instead of

was Answer: Thanks for the reviewer's suggestion. They were corrected in the revised manuscript.

From l. 227 to l. 259: this is a listing of numbers, that are in parts contradictory. It is a bit unclear why you are reporting on all the groups without saying much. Please streamline this. Answer: Thanks for the reviewer's comment. We re-wrote them in a more streamlined way in the revised manuscript.

l. 262 What are qualified reads? Answer: The qualified reads meeting the criteria were listed in the Method section.

l. 263 this means a 99% identity? Answer: Yes, OTUs were clustered at 99% identity.

l. 265 How was the richness determined? Answer: The index of richness, Ace, was determined by the Mothur software (v1.30.1).

l. 272 to instead of with Answer: We revised our manuscript and removed it in our new version.

l. 288 they didn't emerge, they were detected. Also it's not a subset but a clade. Answer: We revised our manuscript and removed it in our new version. pt.

l. 289 this is not remarkable but rather expected, too. Answer: We revised our manuscript and removed it in our new version.

l. 304 I frankly don't believe that nifH is the second most abundant gene anywhere. Answer: Thanks for the reviewer's comment. We revised our manuscript and removed these parts in our new version.

l. 322 ff what you are saying here is that all parameters you included were significantly correlated to the community structure. I am not quite sure how this informs us.

Answer: Thanks for the reviewer's comment. Based on spearman analysis and redundancy analysis newly added to the revised manuscript, all parameters and sampling depth were significantly correlated to the community structure. These results were

consistent with many previous published papers. The new information we provided was: (1) only nutrients were significantly correlated to the community richness, indicating that nutrients were the main factors limiting biomass and within-habitat diversity in the SCS; (2) a depth-dependent distribution pattern was exhibited based on the Redundancy analysis. Depth and environmental parameters might together predict the likely bacterial community in the SCS.

l. 388 How to infer a proportion here? Answer: The proportion here was inferred according to the ratio of the reads of a bacterial group to the total reads of 11 samples.

l. 395 This is most likely because only one group of euryarchaeota is able to fix N2- however, they have never been shown to actively do so in the water column. Thus the statement in line 401 is way too speculative. Answer: We agreed with the reviewer's comment. We revised our manuscript and removed this sentence in our new version.

l. 424 How do you know this? The statistics show you a correlation to the selection of parameters you used in the test. This needs more of a discussion. Also it is somewhat contradicting your conclusion

Answer: Thanks for the reviewer's comments. More discussions were added to the revised manuscript. Briefly, our results showed that the bacterial composition was strongly associated with vertical depth and environmental factors, consistent with previous studies. We hypothesized that the status of nutrients, hydrological parameters, as well as the depth, might could predict bacterial community structure in the SCS. Depletion of nutrients, particularly phosphate, in the upper waters contributed to the low OTU richness of the SCS, that's confirmed by previous studies (Chen et al., 2004; Hwang, 2004), which find that N and P supplements can enhance the Chl a concentration in the SCS. Related information was added to the revised manuscript.

Please also note the supplement to this comment:
https://www.biogeosciences-discuss.net/bg-2018-529/bg-2018-529-AC1-

supplement.pdf

**Fig. 1.** Redundancy analysis ordination of community compositions and environmental variables.

[Figure]

**(a)**

**(b)**

**(c)**

**Fig. 2.** Abundances and distributions of KEGG pathways predicted from PICRUSt: (a) In all sampes; (b) Different depths from SEATS; (c) Different depths from SS1.

[Figure]

**Fig. 3.** Abundances and distributions of KEGG tier 3 KO categories predicted from PICRUSt:
(a) In surface samples; (b) Vertical profiles in SEATS and SS1.

**Supplement:**

**Table 1** Efficient sequencing information statistics of 16S rRNA samples

| Sample | Reads | Mean | Min | Max | OTU | Cover. | Ace | Shannon |
|---|---|---|---|---|---|---|---|---|
| SEATS_surface | 63804 | 432.12 | 387 | 454 | 468 | 99.85 | 517 | 3.50 |
| SEATS_68m | 70929 | 443.36 | 332 | 466 | 394 | 99.83 | 470 | 3.27 |
| SEATS_200m | 77982 | 444.93 | 298 | 462 | 741 | 99.74 | 867 | 3.04 |
| SEATS_750m | 72767 | 446.23 | 278 | 515 | 633 | 99.74 | 756 | 2.72 |
| SS1_surface | 81677 | 430.10 | 291 | 462 | 496 | 99.87 | 555 | 2.63 |
| SS1_105m | 93821 | 429.69 | 338 | 493 | 488 | 99.91 | 530 | 3.29 |
| SS1_200m | 80679 | 435.39 | 366 | 462 | 893 | 99.84 | 956 | 5.03 |
| SS1_750m | 71756 | 436.32 | 305 | 467 | 843 | 99.94 | 857 | 4.78 |
| A2_surface | 61845 | 430.90 | 270 | 464 | 506 | 99.85 | 559 | 3.01 |
| B1_surface | 64261 | 431.38 | 365 | 492 | 479 | 99.83 | 539 | 3.38 |
| C1_surface | 83617 | 430.76 | 347 | 464 | 461 | 99.86 | 545 | 3.02 |

---

## Author Comment (AC2) · 28 Apr 2019

Reviewer 2 comments: This study focuses on the 16S and nifH community structure of the South China Sea. It is a description study with a few sampling points at various depths. The main problem of this paper, is the lack of a narrative. The reader will not be sure what is the main finding of this study against already acquired knowledge. The nifH amplicon approach could be mentioned, but as the nifH primers can only capture a small proportion of the community, the findings could be used in support of other results, rather than forming the main findings narrative. In general, the study is worth publishing as it describes the community snapshot of the SCS. However, this paper needs re-writing in order to better show the importance of the findings.

Answer: Thanks for the reviewer's comments and suggestions. We made extensive changes to this manuscript and re-wrote it according to the reviewer's suggestions. Firstly, the revised manuscript was focused on the community dynamics of the SCS, both the contents related to nifH and nitrogen utilization were removed from our new manuscript. Moreover, redundancy analysis between environmental factors and community structure (Fig. 4) as well as the overall PICRUSt prediction (Fig. 5; Fig. 6) were conducted, and these results were added to the revised manuscript. Secondly, we reconstructed the storyline consisting mainly of "16S rRNA", "environmental influence" and "PICRUSt predictions", in the revised manuscript. Main findings were summarized and discussed with other published studies in the revised manuscript. The importance of our findings lied in the fact that the divergence of bacterial community function and distribution between the nSCS and the sSCS was first reported at the basin scale. Our results again emphasized that bacterial community structures were influenced by environmental factors and provided evidence that the SCS basin exhibited functional zonation among depths which enriched different metabolic potentials.

Introduction: Could you add a few a paragraph about how does the SCS microbiome structure, N-fixation and nutrient limitation etc. compares with other seas. As a reader I would like to know either SCS is an anomaly or does it represent a typical coastal microbiome.

Answer: Thanks for the reviewer's suggestion. We re-wrote the Introduction section in the revised manuscript. New information about the geographical feature, microbiome structure and nutrient limitation of the SCS were added. Comparison with other oceanic regions were also added to the Discussion section. Briefly, the composition of bacterial community was similar with other Pacific Ocean regions with typical oligotrophic characteristics, but exhibited seasonal changes due to the influences of monsoon and eddy.

Results: I find the use of percentage values almost irrelevant considering that you have a single biological replicate per sampling site/depth. I can agree that there is a pattern

BGD
of depth vs surface vs location at the global community level (the whole community pattern as represented on PCoA), but the percentage differences would definetely change with more replications. Please, add that the single replicate does not allow for any statistical analysis to be conducted and instead of using numbers, please just state either some phyla/genera seem to be more/less abundant for a specific sampling point.

Answer: Thanks for the reviewer's comments and suggestions. We agreed that the percentage values used here were irrelevant. Most of the percentage values were removed from the manuscript. A few were retained to show the importance of the species in the survey area. Discussions about the limitation of statistical analysis in present study were added to the Conclusion and Recommend section.

Results: I would like to see a PCA or similar analysis linking nutrient, salinity, temperature with the community structure. Is the influence of the nutrients, salinity etc. smaller or greater than the location? Can you separate them? What I am asking, is the knowledge about salinity and nutrients status of the sampling location enough to predict the likely microbial community structure?

Answer: Thanks for the reviewer's suggestions. We conducted the redundancy analysis between environmental factors and the community structure in the revised manuscript (Fig. 4) according to the reviewer's suggestion. Detailed results and discussion were also added. The results showed that 99.7% of the variance in community structure could be explained by the environmental factors, including nitrate (62.3% of contribution), phosphorus (60.1%), temperature (68.4%), salinity (47.6%), depth (52.4%) in this study. Our results supported the viewpoint that both environment factors and sampling depth were strongly correlated to the bacterial community structure.

(a small remark) please use 100,000 instead of 100, 000 in your sequencing number reports Answer: Corrected.

For the nifH part, please clearly state that no nifH primers are able to provide a comprehensive nifH community profile. Different studies chose different primers. You are BGD
unravelling a part of nifH community. While the comparisons between sites are valid, please remember that this is just a part of the community, and quite likely most of this community is still our of our reach.

Answer: Thanks for the reviewer's suggestions. We deleted it for this part was removed from the revised manuscript.

Figure 1, please add a legend, what depth does the colour signify Answer: Corrected. Please see the attached Fig. 1.

Figure 2a-d please increase the font size of the labels (graph bottoms) Answer: Corrected. Please see the attached Fig. 2.

Figure 3cd plase correct \*Bacteira to Bacteria, please explain, which groups are included in this category in the figure legend. Answer: Corrected. Please see the attached Fig. 3. \*Bacteria meant those unclassified bacterial groups.

**201706CSC sampling station 50 m 100 m 250 m 500 m 20°N 750 m 1000 m SEATS 1250 m 1500 m 2000 m 15°N 2500 m SS1 B1 3000 m 3500 m 4000 m • A2 C1° Ocean Data View 4500 m 5000 m 10°N 5500 m**
6000 m

120°E

115°E

Fig. 1. Sampling sites of the South China Sea basin

110°E

**Fig. 2.** Differences in bacterial community richness, diversity and structure from horizontal and vertical bacterial samples in the Northern and Southern SCS: (a) The richness of the bacterial community; (b) T

BGD
**BGD**